# Adversarial Training of Neural Encoding Models on Population Spike Trains

**Poornima Ramesh**
poornima.ramesh@tum.de

**Mohamad Atayi**
bmeatayi@gmail.com

**Jakob H. Macke**
macke@tum.de

Computational Neuroengineering, Technical University of Munich, Germany

## Abstract

Neural population responses to sensory stimuli can exhibit both nonlinear stimulus-dependence and richly structured shared variability. Here, we show how adversarial training can be used to optimize neural encoding models to capture both the deterministic and stochastic components of neural population data. To account for the discrete nature of neural spike trains, we use and compare gradient estimators for adversarial optimization of neural encoding models. We illustrate our approach on population recordings from primary visual cortex. We show that adding latent noise-sources to a convolutional neural network yields a model which captures both the stimulus-dependence and noise correlations of the population activity.

## 1 Introduction

Neural population activity contains both nonlinear stimulus-dependence and richly structured neural variability. An important challenge for neural encoding models is to generate spike trains that match the statistics of experimentally measured neural population spike trains. Such synthetic spike trains can be used to explore limitations of a model, or as realistic inputs for simulation or stimulation experiments. Most encoding models either focus on modelling the relationship between stimuli and mean-firing rates e.g. [1–3], or on the statistics of correlated variability ('noise correlations'), e.g. [4–6]. They are typically fit with likelihood-based approaches (e.g. maximum likelihood estimation MLE, or variational methods for latent variable models). While this approach is very flexible and powerful, it has mostly been applied to simple models of variability (e.g. Gaussian inputs). Furthermore, MLE-based models are not guaranteed to yield synthetic data that matches the statistics of empirical data, particularly in the presence of latent variables.

Generative adversarial networks (GANs) [7] are an alternative to fitting the parameters of probabilistic models. In adversarial training, the objective is to find parameters which match the statistics of empirical data, using a pair of competing neural networks – a generator and discriminator. The generator maps the distribution of some input random variable onto the empirical data distribution to try and fool the discriminator. The discriminator attempts to classify input data as samples from the true data distribution or from the generator. This approach has been used extensively to produce realistic images [8] and for text generation [9]. Recently, Molano-Mazon et al. [10] trained a generative model of spike trains, and Arakaki et al. [11], rate models of neural populations, using GANs. However, to the best of our knowledge, adversarial training has not yet been used to train spiking models which produce discrete outputs and which aim to capture both the stimulus-dependence of firing rates and shared variability.

33rd Conference on Neural Information Processing Systems (NeurIPS 2019), Vancouver, Canada.

We propose to use conditional GANs [12] for training neural encoding models, as an alternative to likelihood-based approaches. A key difficulty in using GANs for neural population data is the discrete nature of neural spike trains: Adversarial training requires calculation of gradients through the generative model, which is not possible for models with a discrete sampling step, and hence, requires the application of gradient estimators. While many applications of discrete GANs use biased gradient estimators based on the concrete relaxation technique [13], we find that unbiased gradient estimators REINFORCE[14] and REBAR [15] lead to better fitting performance. We demonstrate our approach by fitting a convolutional neural network model with shared noise sources to multi-electrode recordings from V1 [16].

## 2  Methods

We want to train a GAN, conditioned on the visual input, to generate multivariate binary spike counts $y$ which match the statistics of empirical data. We model binary spike trains (i.e. each bin $t$, neuron $n$ and trial $i$ corresponds to an independent draw from a Bernoulli distribution) conditioned on the stimulus $x$ and latent variable $z$ which induces shared variability. We use a convolutional neural network (CNN) $f$ with parameters $\theta$ to capture the mapping from $x$ and $z$ to the firing rate $\lambda$ (Fig. 1), with a sigmoid nonlinearity $\sigma$ in the last layer,

$$\lambda = \sigma(f(z, x, \theta)) \tag{1}$$
$$y|\lambda \sim \text{Bernoulli}(\lambda). \tag{2}$$

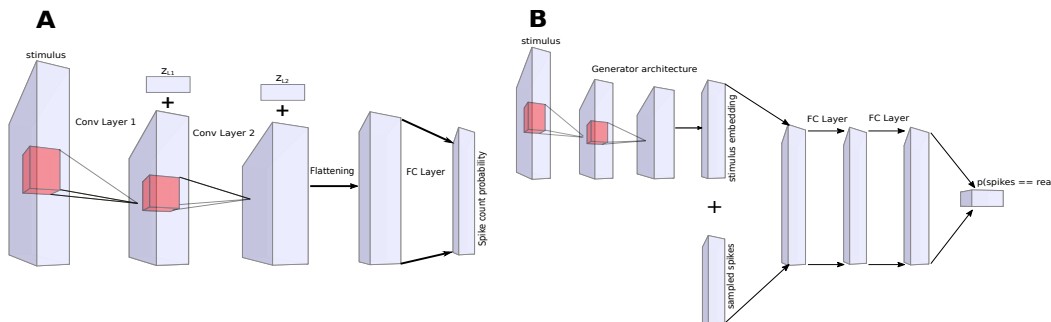

Figure 1: (A) Generator with added noise $z_{L1}$ and $z_{L2}$ (B) Discriminator

The discriminator $\mathcal{D}$ (Fig. 1) receives both the stimulus and the corresponding (simulated or experimental) spike trains. It uses a CNN (similar in architecture to the generator) to embed the stimulus, and combines it with the spike train via fully connected layers. For each timebin and trial, $\mathcal{D}$ outputs the probability of the input spike train being real.

**GAN Training**   The objective is to find the Nash equilibrium of a minmax game between the generator $\mathcal{G}$ and the discriminator $\mathcal{D}$,

$$\min_{\mathcal{G}} \max_{\mathcal{D}} \mathcal{L}(\mathcal{D}, \mathcal{G}) = \mathbb{E}_{y \sim p(y)}[\log \mathcal{D}(y|x)] + \mathbb{E}_{z \sim q(z)}[\log(1 - \mathcal{D}(\mathcal{G}(z)|x)). \tag{3}$$

Due to this objective, GANs are notoriously challenging to train as the training algorithm is sensitive to the gradients with respect to the discriminator parameters. We used the cross-entropy objective as in equation 3, but constrained the discriminator gradients using spectral normalisation [17], and employed gradient norm clipping for the generator gradients.

**Dealing with discrete data**   Obtaining the gradients for the generator requires backpropagation through both generator and discriminator networks. Most applications of GANs have been on continuous data. However, spikes are discrete and thus, the generator has a discrete sampling step which blocks backpropagation. Previous attempts to overcome this problem include using concrete relaxation [13], REINFORCE [14] or REBAR [15] to estimate gradients. Concrete relaxation approximates the binary variables as continuous values which are close to 0 and 1. This allows backpropagation through the sampling step, but leads to biased gradients. The REINFORCE gradient estimator provides unbiased but high-variance gradients using the log-derivative trick. The REBAR

gradient estimator combines concrete relaxation and REINFORCE by using the relaxed outputs as a control variate to the REINFORCE gradient. For the applications and hyper-parameter settings we used, we found that both REINFORCE and REBAR performed better than concrete relaxation, and that REBAR did not have much improvement over REINFORCE (Fig. 2).

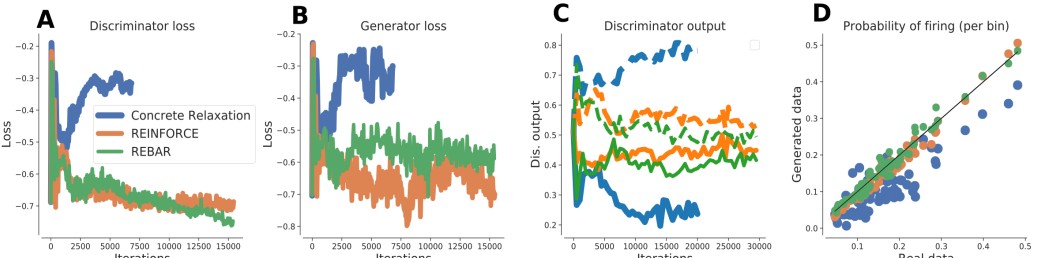

Figure 2: Comparison of different gradient estimation methods for GAN on experimental data. (A) Discriminator loss for concrete relaxation (blue), REINFORCE (orange) and REBAR (green) across training epochs. (B) Generator loss for the 3 gradient estimation methods across training epochs. (C) Discriminator output for real data (dashed lines) and generated data (solid lines) from GANs trained with the 3 gradient estimation methods. (D) Firing rate probability (per bin) calculated on output from GAN generators trained with 3 gradient estimation methods.

**Architecture and dataset** We fit the GANs to a publicly available dataset [16], consisting of 69 cells recorded from macaque V1, while animals watched a 30s movie repeated 120 times. The movie consisted of 750 frames of size 320 x 320 pixels, which we downsampled to 27 x 27 pixels. We binned the spikes at a 40ms resolution, and binarized the resulting spike trains. Since only 5% of the spike counts in each bin were non-binary after re-binning, we assumed that the binarization would not significantly alter the results.

For the generator, which receives 10 consecutive movie frames of size 27 x 27 pixels as input, we used a 3-layer CNN architecture similar to Kindel et al. [18] (Fig. 1A) – layers 1+2: convolutional with 16 and 32 filters, size 7 by 7, each followed by a MaxPool layer with kernel size 3 and stride 2, followed by LeakyRELUs with slope 0.2. The final layer of the CNN was a fully connected layer with units equal to the number of neurons in the dataset. To capture the stimulus-independent variability shared between the neurons, we added Gaussian white noise to the units of the convolutional layers. The noise was shared between the units of these layers, multiplied by a separate weight for each unit. The discriminator network consisted of a CNN embedding for the stimulus, similar in structure to the generator, but without the added shared noise (Fig. 1B), and 5 fully connected ReLU layers.

**Training** We trained the two networks in parallel for 15k epochs, each consisting of 2 discriminator and 1 generator update. With batch size 50, we used ADAM with learning rate 0.0001, $\beta_1 = 0.9$ and $\beta_2 = 0.999$ to optimise the network parameters. The first 650 timebins were used for training the networks and the last 100 timebins for validation. All hyper-parameters were set by hand.

## 3 Results

We fit a 3-layer CNN generative model of binary spike counts to neural population data recorded in the V1 area of macaque visual cortex [16], using adversarial training as described above. For comparison, we fit a CNN with a similar architecture to the GAN generator – but without the shared noise layers – to the same dataset, using supervised learning, i.e. by optimizing the cross-entropy between predicted firing probabilities and experimentally observed spike trains. We also fit a Dichotomised Gaussian (DG) model [5, 19], which explicitly represents shared variability via the covariance matrix of a multi-variate Gaussian distribution.

On the training data, all approaches reproduced the gross structure in the spike train rasters (Fig. 3A) and accurately captured the firing rates (here: spike-probabilities per bin, Fig. 3B left). However, the supervised model did not accurately reproduce total pairwise correlations between the neurons (Fig. 3B center), since its noise-correlations 3C) are constrained to be 0. Thus, the histogram of population spike counts for data generated from the supervised model is also substantially different from that of

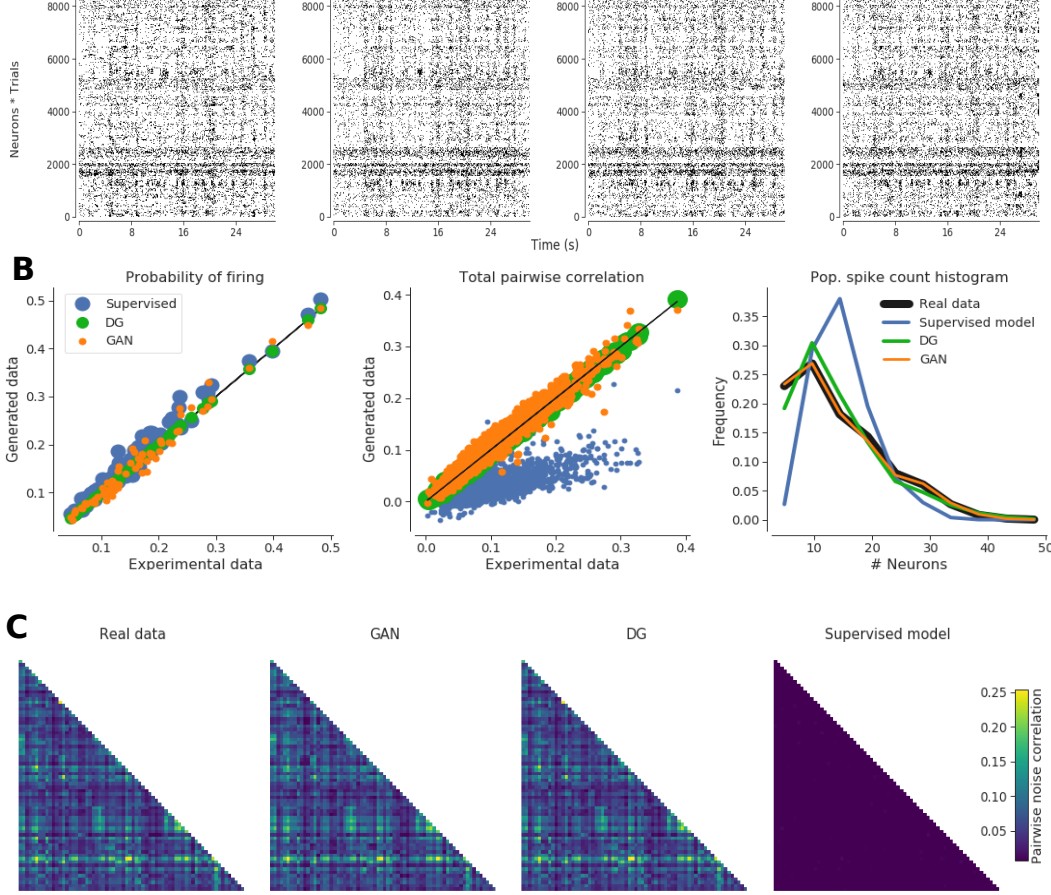

Figure 3: Model comparison on experimental training dataset. (A) Spike train rasters for data and models. (B) Firing rates (per bin, left) and total pairwise correlation (middle) of supervised (blue), DG (green) and GAN generator (orange) model versus data. Right: population spike count histogram for data (black), supervised (blue), DG (green) and GAN generator (orange) model. (C) Pairwise noise correlation matrix for data and models.

the real data (Fig. 3B right). The DG model accurately captured the total correlations and the pairwise noise-correlation matrix since it designed to match the peri-stimulus time histogram (PSTH) and the noise-correlations of the spike trains. However, it did not perfectly capture the population spike count histogram, as it only models second order correlation between neurons, while the population spike count histogram also depends on higher order correlations. In contrast, the GAN generator, with the addition of just a few shared noise parameters to the supervised model, was able to accurately capture the total correlation, the population spike histogram and pairwise noise-correlation matrix.

However, we found that neither model generalised well to the held-out test-dataset, possibly because of the short training-set and high variability of this dataset [16]. When we fit a CNN with the exact same architecture to simulated data with higher SNR and the same dimensions as the V1 dataset, we found that the CNN was able to capture the PSTH and SNR in the test data (Fig. 4).

On the V1 dataset, the adversarially trained CNN and the DG model were similarly good in reproducing correlations and spike count histograms. This might occur if the spike trains of the neural population are homoscedastic (i.e. the variability does not depend on time or stimuli), as assumed by the DG model. Adversarial training is limited only by the flexibility of the network, and can also capture heteroscedastic, i.e. stimulus-dependent noise. Hence, we simulated heteroscedastic data with the same dimensionality as the V1 dataset. We added latent noise to the second layer of the CNN, with variance proportional to the squared input from the previous layer. We fit both a CNN and a DG model to this simulated dataset. We found that CNN trained with the GAN framework was

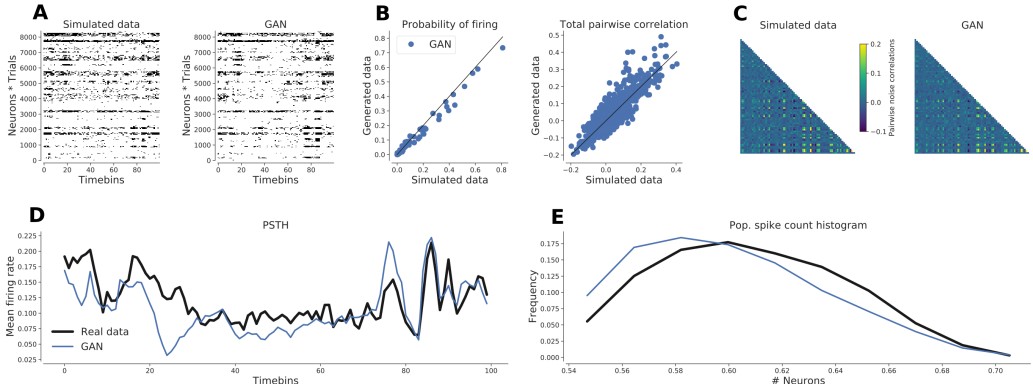

Figure 4: GAN generator fit on simulated test dataset. (A) Spike train rasters for test data and model. (B) Firing rates (per bin, left) and total pairwise correlation (right) of GAN generator model versus data. (C) Pairwise noise correlation matrix for data and model. (D) PSTH averaged across trials and neural population for data(black) and model(blue). (E) Population spike count histogram for data (black) and model (blue).

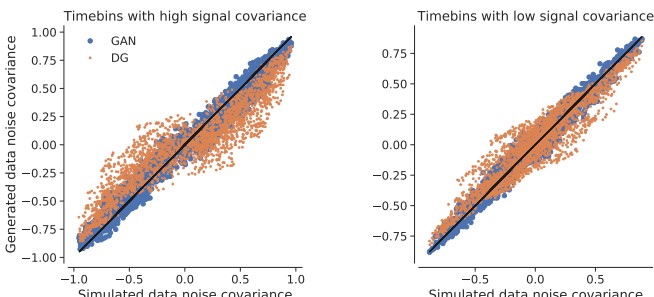

Figure 5: Pairwise noise covariance for simulated heteroscedastic dataset Noise covariance for simulated data versus GAN (blue) and DG (orange) models in high signal covariance timebins(left) and low signal covariance timebins (right).

able to accurately capture the covariances estimated from 'low signal' and 'high signal' timebins separately (Fig. 5), unlike the DG model.

# 4 Discussion

We here showed how adversarial training of conditional generative models that produce discrete outputs (i.e. neural spike trains) can be used to generate data that matches the distribution of spike trains recorded in-vivo, and in particular, its firing rates and correlations. We used unbiased gradient estimators to train conditional GANs on discrete spike trains and spectral normalisation to stabilise training. However, training of discrete GANs remains sensitive to the architecture of the discriminator, as well as hyper-parameter settings. We showed that we are able to successfully train adversarial models in cases where supervised and Dichotomised Gaussian models fail.

In future, adversarial training could be used to capture higher-order structure in neural data, and could be combined with discriminators that target certain statistics of the data that might be of particular interest, in a spirit similar to maximum entropy models [4]. Similarly, this approach could also be extended to capture temporal features in neural population data [20] such as spike-history dependence or adaptation effects. Since we condition the discriminator on the input stimulus, adversarial training could be used for transfer learning across multiple datasets. Generative models trained this way to produce realistic spike trains to various input stimuli, may be used to probe the range of spiking behaviour in a neural population under different kinds of stimulus or noise perturbations.

## Acknowledgements

We thank all members of CNE, especially David Greenberg and Artur Speiser, for feedback and discussions. We thank Matthew Smith and Adam Kohn for sharing data via CRCNS.org. Funding was provided by the DFG through SFB 1233 (276693517).

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
