# OpenReview forum: "Adversarial Training of Neural Encoding Models on Population Spike Trains"
_NeurIPS.cc/2019/Workshop/Neuro_AI — Real Neurons & Hidden Units @ NeurIPS 2019 Oral_

### Official Review · AnonReviewer3 · 2019-09-26
**Useful method for both Neuro and AI**

**Clarity:** 4

**Comment:**

The authors use GANs to generate spike trains that match experimental data. Features of the data that are compared are 1) firing rates, 2) pairwise correlations, 3) population spike count histogram. The authors show that, on the training data, GANs can reproduce all features of the data very well. The authors also show that GANs outperform a dichotomized Gaussian model when fitting the population spike count histogram (although the difference is small).

Areas for improvement: It might be informative to also show validation results and not only training data results. On a similar note, the next step of assessing transfer learning across data sets would be very interesting.

**Category:**

Common question to both AI & Neuro

**Clarity Comment:**

The paper is very well written and details are appropriately explained.

**Evaluation:**

4: Very good

**Importance:**

4: Very important

**Importance Comment:**

The authors use GANs, as an alternative to likelihood based approaches, to generate spike trains that match experimental data. Generating realistic spike trains is useful for both Neuro and AI.

**Intersection:**

4: High

**Intersection Comment:**

Generating realistic spike trains is useful for both Neuro and AI applications.

**Rigor Comment:**

I defer to other reviewers on the GAN training procedure as I have limited expertise in the area.

**Technical Rigor:**

3: Convincing

---

### Official Review · AnonReviewer1 · 2019-09-26
**An interesting new approach to study an old problem**

**Clarity:** 4

**Comment:**

Overall, I think this represents an interesting direction. Although the results are preliminary, it does look promising.  I have several concerns/suggestions:
1. it would be helpful if the authors could discuss the applications of the method in the context of a neuroscience question, (maybe the even showing one real application)?
2.  it would be useful if the authors could quantify the high-order correlation structure. By looking at the rasters in Fig. 2A, although it seems GAN-based models looks visually more similar to the real data comparing to others, to make the argument precise,  it would be good to quantify it.
3. The improvement of GAN-based models over DG seems to be not totally convincing.



**Category:**

AI->Neuro

**Clarity Comment:**

The writing is clear.

**Evaluation:**

3: Good

**Importance:**

3: Important

**Importance Comment:**


A basic question in system and computational neuroscience is to come out with good  models of neural spike trains. This paper introduces a new method to capture the response patterns of population spike trains, by using Generative adversarial networks (GANs).

**Intersection:**

3: Medium

**Intersection Comment:**

This paper uses a popular method in AI to address a classic neuroscience problem.

**Rigor Comment:**

The techniques presented in the paper appears to be solid and convincing. The authors compared the proposed method with previously proposed models, i.e., dichotomised Gaussian model,  supervised model.   The authors found that GAN-based model is overall a good model. It can capture the probability of firing rate, pairwise correlation, spike count histogram, and the rasters.  Other methods can fail in one or several of these aspects.

**Technical Rigor:**

3: Convincing

---

### Decision · Program_Chairs · 2019-10-02

Accept (Oral)